# Exploring Other-Than-Human Identity: Religious Experiences in the Life-Story of a Machinekin

Stephanie C. Shea 

Faculty of Humanities, University of Amsterdam, 1000 BP Amsterdam, The Netherlands;
stephanie.shea72@gmail.com

**Abstract:** The term *Machinekin* denotes a sub-group of a larger Internet subculture known as Otherkin: while recognizing they have a human body and mind, these people nevertheless identify as being other-than-human. Machinekin therefore identify as a machine of some sort. In attempting to study this subculture, qualitative psychological research methods are used, combined with digital ethnography. Postmodern theories of identity formation, such as narrative identity, and especially McAdams's seven features of the life-story, are implemented in order to interpret how Neve, a Machinekin, came to understand his non-human identity, as well as the role religion has played in his identity configuration. Additionally, the function of religion as it applies to finding meaning in conflicting circumstances is also considered. Neve's experiences can be seen as an example of how religion and identity are interrelated, with the story showing what key events led Neve to look to religion for answers to difficult questions that arose in his early years. The interpretation of these events eventually led to an understanding of Neve's lived experiences, and to a sense of purpose for his life. It also demonstrates how Machinekin challenge attitudes surrounding identity and the boundaries of what constitutes a "person".

**Keywords:** narrative identity; life-story; other-than-human identities; conflict; boundary crossing; meaning-making; anomalous religious experiences

## 1. Introduction

The discussion of "experience" as it relates to religion is reappearing into focus as an important area of study, especially as it relates to personal identity. Contemporary accounts of unusual experiences are diverse, such as hearing voices (Luhrmann 2012) or sensations of travelling out of one's body, and can be "deemed religious" by the experiencer (Taves 2009, pp. 22–26). Conveying an experience most often occurs through interpretation, especially by telling a story about it to others. These powerful, influential events make an impression on a person and can shape a person's sense of personal identity, as it is generally accepted that what happens to a person can have an influence on who they become. In recent decades, identity has become a somewhat controversial social topic with regard to gender and sexual orientation, but also with regard to those who claim other-than-human identities, one such group being called "Otherkin." This subculture is currently found predominately online, but has its origins in older, pre-Internet worldviews. However, the location of the Internet is important as it is where much of the current research is done regarding the worldwide (micro)communities now in existence.

Otherkin are people who, while recognizing they have a biologically human body, do not feel they are completely human. The other-than-human identification, that is specifically stated (by themselves) as *not* being chosen by the person, can be expressed in diverse types and combinations: animals; plants; mythical beings; non-corporeal entities; aliens; A.I. forms; and recently, machines, to name but a few (Shea 2019, pp. 4, 5). "Otherkin," therefore, is the umbrella term that is used for this

conglomeration of types. Other-than-human expressions have a long history that can be found, for example, in archaeological and anthropological reports of animals and humans melding or blending together (Antl-Weiser 2018; Lindstrøm 2012); the philosopher Diogenes of Sinope and Cynicism, with the nickname "cynic" meaning "doglike" (Branham 1998, 2009); Nordic literary epics that speak of elven races that interact with humans (Hall 2007; Johannsen 2016); numerous tales about werewolves found in Greek, Roman, Jewish, French, and Irish sources (Bourgault du Coudray 2003; Gordon 1974; Mullin 1999; Nelson 2012; Panxhi 2015; Schwartz 1987; Shyovitz 2014); and in accounts surrounding vampires that span from the late sixteenth century to the present day (Bahna 2015; Introvigne 2001; Melton 1999; Nelson 2012; Laycock 2010). Otherkin in its current, contemporary form is said (emically) to have arisen out of American counter-culture movements of the 1960s and 1970s, with Otherkin (including those who considered themselves "real" elves, vampires, werewolves, etc.) emerging in two general "movements"—the first in the 1970s; the second in the 1990s with the introduction of the Internet and the World Wide Web (Scribner 2012, pp. 5, 10, 12–16, 19, 21, 26).

Current research—although still limited, as Otherkin is a relatively new area of research in academia—expresses diverging opinions as to what Otherkin represent. Literature from scholars in the field of Religious Studies (Davidsen 2013, 2014; Kirby 2012, 2013; Laycock 2010, 2012; Robertson 2012, 2013) appear to place Otherkin into a type of functionalist religious or spiritual social framework, with the exception of Kirby (2013), who proposes a substantive definition of religion, even though they all verify that Otherkin themselves do not claim to be a religion. Ultimately, while both definitions of religion could be acceptable *in theory*, the attempt to categorize Otherkin as a religion creates tension with Otherkin self-perception, as they emphasize they are not a religion at all. Therefore, this author questions whether Otherkin identity, as a whole, should be seen as fitting into the categories of religion or spirituality. This author does not claim that religion or spirituality is absent from Otherkin narratives; it certainly can play a role with regard to the worldview of an individual (as will later be demonstrated), but it is not the determining factor for *being* Otherkin, or "Otherkinity," as such.

This author's own research into the Otherkin community for the past six years conflicts to a certain degree with the perspectives of the academic literature stemming from the area of Religious Studies. While the scope of these conflicts cannot be listed here in full, as it is beyond the scope of this article, one major difference can be mentioned. While the scholars primarily focus on Otherkin in terms of function and performance, this author's research data suggest that Otherkin are a philosophical ontology of *being* other-than-human that is separate from any religious constructions, and is seen (by Otherkin themselves) as being an epistemic realization, present from birth, and comparable to sexual orientation (Shea 2019, pp. 45, 53, 55, 58). This stance is more in agreement with the ontological position that Proctor (2018, p. 489) promotes.

Furthermore, as Otherkin identity is not easily explained by religious or spiritual frameworks, the focus of research could be shifted to that of identity, including the question of how experiences deemed religious can play a role in the realization of a person's sense of who they are. This author's research has shown that precisely due to the diversity and complexity of Otherkin identity types, qualitative research methods (such as participant-based observation and interviews) should be implemented in order to gain a better understanding of the people within this group. The researcher must also invest time in communication, e.g., performing longitudinal studies, with Otherkin in order to gain a better understanding of the ideas discussed in the various (micro)communities, and to establish trust with the research participants (Shea 2019, pp. 42, 43, 59, 65, 66). By doing this, the researcher can then determine how, in some cases, religion *can* play a role for certain Otherkin as it relates to the realization of their other-than-human identity. One way of determining this is by the use of McAdams's *narrative identity* (McAdams and McLean 2013) and the *life-story* (McAdams 1997).

## 2. Study Aims

This article will focus on the case study of a Machinekin (i.e., a type of Otherkin; these individuals identify as a machine of some kind) called Neve, who identifies specifically as an AMS Neve VR52

music mixing console (see Figure 1 below). His "Awakening" narrative (i.e., how he came to realize and understand his other-than-human identity), together with his "life-story" interview, will be used to illustrate McAdams's life-story framework. McAdams discusses how each person's narrative includes seven features that illustrate the attempt of the self to create a sense of unity (of different aspects of one's identity) and purpose; these features are:

1.  *Narrative tone* (overall emotional nature of story),
2.  *Imagery* (creative use of language to express a special quality of experience),
3.  *Theme* (autonomy and communion with the environment),
4.  *Ideological setting* (religious, ethical, or political beliefs and/or values),
5.  *Nuclear episodes* (crucial or pivotal moments that contribute to the development of a person),
6.  *Imagoes* (aspects or models that represent a good life),
7.  *Endings: The Generativity Script* (endings or new beginnings that leave the person feeling fortunate and benevolent) (McAdams 1997, pp. 65–71).

McAdams (1997, pp. 47–49) discusses how Robert J. Lifton's postmodern theory of the "protean self" demonstrates that a person's sense of identity, or "who they are", is comprised of many "personas" (i.e., "wife," "teacher," etc.) that is flexible in nature. The narrative "life-story" is a flexible construct that reveals aspects such as race, gender, and class position as these can unfold over time. These "self-conceptions," taken together, form a type of "pattern" that offer a sense of meaning and direction to a person. (McAdams 1997, pp. 56–63). By using this construct in combination with the qualitative psychological methods listed below, this study attempts to interpret how Neve came to understand his unique experiences, recognize his other-than-human identity, as well as the role religion has played in his identity. For the ease of the reader, the life-story characteristics will be presented, as much as possible, to correspond chronologically with Neve's narrative.

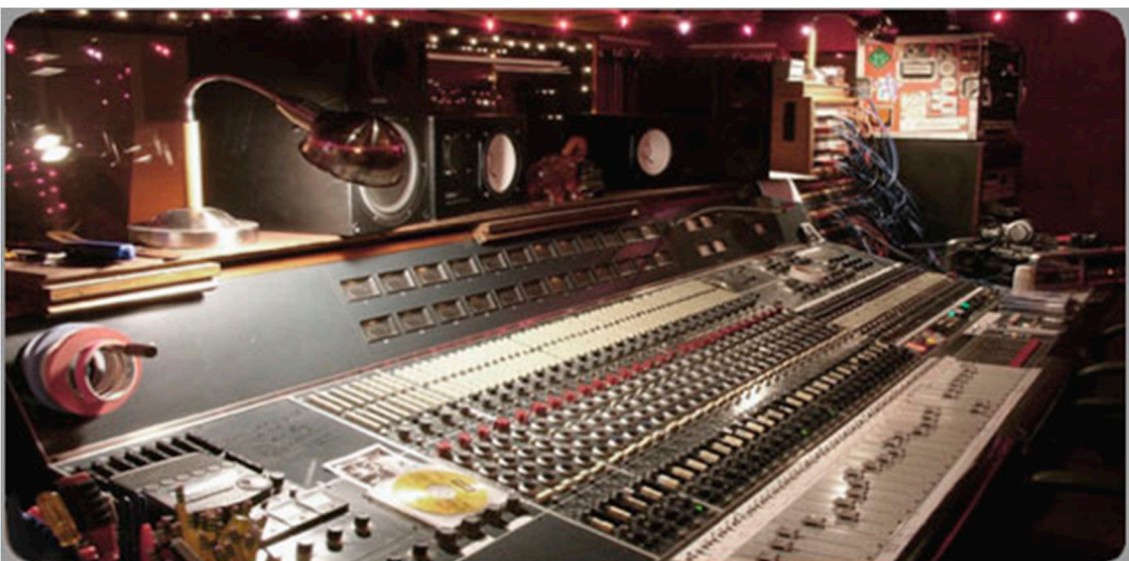

**Figure 1.** A Neve mixing console. Image (Sound City Studios 2013).

## 3. Methods and Interview Information

In order to attempt to answer the question of how the conviction of other-than-human identity arises, and how an Otherkin experiences this subjective, personal identity, alternative approaches taken from the traditions of phenomenology and hermeneutics could be utilized. As the research generally acknowledges, each Otherkin has a unique narrative of the "Awakening" experience that explains how he/she came to know of his/her other-than-human identity (Shea 2019, pp. 5, 10, 11, 17, 18, 21, 68–73). This subjective experience is at odds with a positivist view of what is "real," yet

to fully understand this experience, a first-person viewpoint of the "life-world" of the individual is needed, and therefore bracketing, or *epoché*, on the part of the researcher is required in order to focus on the phenomenon itself (Ashworth 2015, pp. 10–12). Qualitative psychological methods that can be of interest for this purpose are: Smith and Osborn's interpretative phenomenological analysis (IPA), that explores "personal meaning and lived experience"(Smith and Osborn 2015, p. 25); Kozinets's method of netnography, i.e., digital ethnography that combines grounded theory from Strauss and Corbin (1990); hermeneutic coding from Arnold and Fisher and Thompson et al. (1994); and Hine's virtual ethnography (2000) together with participant-based online fieldwork (Kozinets 2010, pp. 60, 119, 120). These methods can be combined with postmodern theories such as narrative psychology as demonstrated in the ideas of Ricoeur, who emphasizes narrative as a means to understand one's self or identity within a constant state of change (Murray 2015, pp. 85–87), as well as narrative identity (McAdams and McLean 2013) and the life-story (McAdams 1997), mentioned above. The latter two methods focus on the ever-changing and evolving story that a person embodies in the attempt to afford meaning, "unity and purpose" to an individual's life (McAdams and McLean 2013, p. 233); McAdam's life-story, in particular, is a type of construct that contains seven related features that show how the self creates an identity by using a narrative (McAdams 1997, pp. 65–71). This will be discussed in more detail below.

As listed above, the seven features of the life-story are analyzed by means of a "toolkit" that allows for bracketing; participant-based online fieldwork, including semi-structured interviews; textual analysis and line-by-line coding. Coding produces themes that can assist in interpreting the experiences and perspectives of the participant (Gibbs 2011, pp. 54, 71).

This author's communication with Neve (age 31) began in 2016 by way of his member page (or blog) on an artistic social media network website. Through private, textual communication (over several days) on this site, Neve shared his "Awakening" story with this author. At this time, Neve was not active in the online Otherkin communities. After our initial communication, Neve returned to various online micro-communities, and also created his own public website along with a private, invitation-only Discord server. In 2018, Neve agreed to participate in a longer, semi-structured interview, although he was not aware that it would be a life-story; this was intentional, as this author intended to keep the interaction as spontaneous as possible. This communication took place over several days, and was realized through written communication, as Neve preferred this over face-to-face conversation. The first semi-structured interview was in real time, lasting for almost two hours. Due to internet connection issues in Neve's area, the subsequent days were used for completion of the life-story as well as for follow-up questions. The interview was conducted on Neve's private Discord server via the private messaging function. This author requested Neve's permission to use his information before writing this article; this was granted through text communication. Additionally, Neve reviewed the finished manuscript, and had no objections for the article to be submitted for peer-review and publishing.

Information taken from both communication sessions (i.e., the "Awakening" story and the "life-story," that will be presented as the "narrative" in the next section) have been used to provide the data presented in this article. This decision was made due to the fact that Neve did not provide any information regarding his Awakening experience in the life-story interview, as this was discussed in detail a few years prior, and would have been unnecessarily repetitive. Combining the two narratives is important for context, and provides a more comprehensive picture of Neve's life experiences. Manual line-by-line coding, with one-degree axial coding, was performed on the transcripts; this coding provided a list of themes to use for interpretation of the material. This is in agreement with the processes (coding, noting, abstracting, and comparing) as put forth by Kozinets (2010, p. 119), as well as the function of coding as a "shorthand reference" that allows a researcher not only to interpret the participant's experiences, but also to give a "voice" to the participant in order to understand how he or she experiences life over time (Gibbs 2011, pp. 54, 71).

## 4. Analysis

### 4.1. Nuclear Episodes

There are many important, key moments in Neve's narrative. He was born prematurely, and nearly died due to his umbilical cord being wrapped several times around his neck. These complications left him with high-functioning autism; in addition to this, he was also born with mild cerebral palsy that affected his eyesight. One of his earliest memories (approximately age four years old) is being "incredibly focused on the car stereo" while his mother was driving, listening to music from Reba McEntire. Later, at ages six and eight, he "encountered" music mixing consoles and sound systems at school events; these machines left a deep impression on him, and he was eager to learn more about these machines. At age ten, he recounts the death of his best friend and her family who died in a house explosion; Neve had been invited to sleep over that evening, but his mother would not allow it, having claimed a sort of premonition that something bad might happen. Neve felt it was a miracle that he was saved from death, but the event (along with the September 11 attacks in 2001) left him questioning why God let horrible things happen, and this was the start of his search for answers. Another crucial event occurred at age eleven—a dream

> *more vivid and real than any I had ever had. I was never one to remember my dreams, but I still remember every detail of the one I had that night. I was the real me in that dream. It felt more comfortable than I had ever been, warm and content. Upon waking, I was breathless. It all made sense to me.*

Neve states that these events were "all an integral part to the awakening process." While he did not know that there were others like him called "Otherkin," he states that he "had figured out his internal identity" by this time. While in middle school (approximately age twelve), his beliefs were "solidified" when he saw a picture of an AMS Neve VR52 mixing console on the internet. He shares,

> *I stared at the image for hours because it was me from my dream staring back at me through the old CRT computer screen. [...] The fact that it matched what I saw in myself was what assured me that perhaps all of my feelings weren't just an autistic person's obsessive nature gone too far. There was something deeper to it... beyond the reaches of a malfunctioning human brain.*

Yet another pivotal moment came at age sixteen: Neve was operating the sound system at a megachurch he attended, and told someone he was Machinekin. He was also suffering with gender issues at this time. He states, "I was kicked out for being who I am. It broke my heart, and my faith." During this same time period, Neve states he was "in a very dark place" due to a "bad home life [with his biological father] and depression." At age twenty, Neve states he realized "God's calling" for him to move away to study at a liberal Catholic seminary. He began hormone treatment therapy at this time and legally changed his name. When he returned home, however, his biological father disowned him for being transgender.

### 4.2. Narrative Tone

Neve's life has been marked by adversity and suffering yet his tone remains hopeful and determined. While he has had to contend with stressful issues surrounding his autism, his transgender identity and Machinekin identity (such as bullying, and rejection from some family and friends), poverty, and depression throughout much of his life, he holds an attitude of perseverance and thoughtfulness. He does often refer to God's agency in his life, and a sense of destiny, even if he does not have all the answers to his existential questions. With regard to his autism and God's agency in his life, he states,

> *Autistic people think like machines. Our perspective of the world around us is very mechanical, black and white, one and zero. The question I've had to ask myself is: are the reasons I feel I am machinekin the same reasons I was diagnosed autistic? If they are, am I only just autistic? Or am I a machine that*

*thinks like a machine, and humans labeled me autistic because of it? [...] I don't think autism is the root of all my identity, rather the opposite. The conditions of my identity have given me an autistic brain because that's the brain that would work best for my soul. [...] There's one thing I greatly value that I didn't have as my inner self, and that's freedom. [...] I get to know the outside world beyond a soundproof control room. It is a gift that I have been given by a loving Creator.*

### 4.3. Theme

Two aspects of this feature are "agency and communion," i.e., how one tries to be autonomous and in control over his/her life, and how one connects and interacts with his/her environment. One observes how these relate to each other as they appear in the narrative (McAdams 1997, pp. 66, 67). When looking at Neve's narrative regarding his connection to his mother, siblings, and maternal grandparents during his early teenage years, it is clear he was supported by his mother's family once he fully realized his machine identity: "I told my entire family like I had just discovered the greatest news. [...] To this day, they all refer to me as a machine." Neve also began exploring the online Otherkin community at this time (early 2000s). He states, "One night, I decided to search "I do not feel human." This led me to a forum for Otherkin. The very first community I joined welcomed me." As he sought out other website forums, however, he encountered increasing negativity from others, this mainly due to the general sentiment by many Otherkin at that time that machine *objects* could not be Otherkin. He admits that he was young, and not able to express himself adequately; this led others to believe he was a "troll" (someone who deliberately enters online communities with the intention to be disruptive and offensive), and he was banned. This happened repeatedly, and Neve eventually "gave up" trying. He states that

*a few years later I found myself longing again to know others. All I found was hate and discrimination,* [Machinekin were] *pushed out of the community by force. It saddened me...I wanted to reach out to those people cast aside.*

As Neve matured into adulthood, he also struggled to be in control of his life due to an array of factors, such as poor health, and conflicts with his biological father. He also had trouble finding and keeping jobs due to his autism, that created financial issues for him. Yet it is clear from his narrative that Neve wanted to connect with, and help, other people. Referring to his time in seminary, living with eight others in a crumbling boarding house, he speaks with a sense of optimism:

*Everyone there had each other's backs. We had to, because none of us would have gotten by without the others. They fed me when I had no food, and I fed them. I learned more about God, people, and the world in that place than I did in any of my formal seminary training. [...] Among the people I lived with, I learned that God is truly in everything. There is light even in the darkest place.*

### 4.4. Ideological Setting

When looking at Neve's beliefs and values (whether they be religious, political, etc.), how they came to be, and how he confirmed what he feels is "right" for his life, one sees that Neve is quite steadfast in his viewpoints, even if a few areas are not completely clear to him. His mother attended church, but Neve says he did not grow up in a religious home, as his mother "taught that it was important [for her children] to find our own paths." While Neve was designated as a female at birth, he says he felt a strong calling in his teenage years to serve God as a minister, yet was told by his Protestant Christian pastor that such a thing was impossible, as "women aren't supposed to lead in the church." This rejection angered Neve greatly, and as a result of this, Neve left this church and went on to experiment with various religious systems (including Wicca, Theistic Satanism, Buddhism, and Pentecostal Christianity) before finding his own path in liberal Gnostic Catholicism. Neve's basic belief regarding human nature is that "people come in all kinds of broken, and broken doesn't mean bad. People genuinely are inherently good." It is also clear that he feels his own existence is a miracle. This could also be included under the *theme* feature, as Neve explicitly states that, as a baby,

*I spent some time in the hospital, but miraculously lived. "But miraculously lived"—that's kind of the whole theme of my story and also the roots of my religious beliefs.*

Neve emphatically states that he believes in God, and as an ordained priest in an autocephalous, liberal Catholic jurisdiction, he firmly believes that "God made me this way." He expands on this, saying,

*I don't know why God made me as he did, but being machinekin has been an unwavering part of my identity since I was a young child. [...] A part of me likes to think that God put me through my traumatic birth, gave me autism so that my mechanical soul could better mesh with my organic brain. My faith and my beliefs about my identity are very strongly intertwined. [...] It should be stated that I am a staunch animist. I believe that everything is capable of having a soul, and I have my whole life. [...] I can't explain the exact workings behind all of this. I don't know exactly how I came to be an AMS Neve VR52 in a human body. I can't give some long, theological nor scientific explanation of it all. I only know that it's true to me. Was it reincarnation? Or was I just created this way? Or is it merely a product of my autism? The truth is, I can't answer those questions. Nobody can. [...] It's all a part of my identity that has been there most of my life. I am a machine, that is all I know for certain. I have memories of being my true self, more concrete than those of this current existence.*

On the subject of animism, Neve speaks in detail about how he feels that machines can have souls:

*We're all made of the same elements, we call came from the same place. What makes the chemicals this body is comprised of any more worthy of a soul than the ones a mixing console is comprised of? Why can't a machine have a soul? It is because a machine is incapable of expressing 'life'? There are humans who can't speak, cannot move, cannot even think for themselves, yet we consider them 'alive.' Bodies are just a bunch of chemical reactions. It's no different for machines. I believe very strongly in the existence of spiritual energy. I can feel it all around me. It flows through every atom in existence.*

Furthermore, he expresses his ideas about why he feels that being a machine object, such as a mixing board, is no different from other machines, such as robots, that are considered by the majority of the Otherkin community to be legitimate Machinekin:

*I don't think there's much of a difference at all between a robot and a mixing console when it comes to its soul. A robot is just as much of a machine, an object built by humans. Because it walks on two legs and is programmed by humans to behave like humans doesn't make it any less of a machine.*

### 4.5. Imagery

Neve is generally straightforward and matter-of-fact when communicating; however, he also uses quite poetic and vivid language when expressing himself, often with a sense of wry wit. For example, when speaking of his youth, he shares,

*I grew up with three siblings. We used to play house, like any other little kids; [...] instead of being the dad, or the knight in shining armor saving my sisters from an evil king, I chose to be the stereo system. [...] I had always felt weird in my skin. This small, smooshy [a portmanteau of 'smoosh' and 'mushy'] body that runs on strange things and does even stranger [things] has never felt like home to me, and when people would make any sort of comment regarding my physical body, I would get so frustrated that they couldn't see me as I saw me. In my mind, I'm a 2000-pound machine that has to turn sideways through double doorways.*

When speaking about his time at seminary, he describes the state of the cats in the dangerous, drug- and crime-ridden neighborhood he lived in during this time period as looking "like they'd been run through the workings of a vehicle too many times." Later, when he was without a home, he temporarily lived with friends. Of this time, he recollects that he

*ended up on a shitty leather couch in a hoarder house. I was there for eight months, and I can tell you that soap, and water, and cleaner, and vacuums, and taking out the garbage are very wonderful things that should be pursued in life. No matter if I had just showered, just put on clean clothes, I stunk.*

When asked about his ideas and feelings regarding machines, and how this corresponds to his own faith, he shares,

*Everything is beautifully and wonderfully crafted by the Almighty, including machines. In this way, God made man, and manmade machines. Therefore, machines are also of God. They're even in the Bible: the great mechanical chariots that were ridden by angels in the book of Ezekiel, the war engines of Revelations, all were holy, ordained machines made by God. Another part of liberal theology involves reincarnation, but I don't think mine was a mistake at the soul factory. I'm here in this body for a purpose and a reason.*

### 4.6. Imagoes

Neve shares how his early upbringing with his mother and his maternal grandparents had instilled important values within him regarding how one should live his or her life:

*I was raised to be true to myself, and my family did more than just preach that. They practiced it constantly in their everyday lives. I come from a liberal background where it's more than alright to be gay, it's good to have quirks, and a person's internal identity is their own to be discovered and should be respected. Others, unfortunately, were not so accepting and understanding. I used to tell everyone [of his identity] when I was a teenager. It has cost me friends, jobs, and mates. [...] It's all about discretion. [...] I think it's important to share with those that have become my loved ones so they may have a better understanding of me and why I do the things I do.*

The examples of "right living" helped Neve to present himself to his peers in a way that felt authentic to him, although not without issues in his peer group, or with employers:

*I used to carry about 40 pounds of [ audio/recording] gear on me at all times, and I dressed in clothes that gave my body an even more square and boxy appearance on purpose. These were things I did because they made me feel comfortable in my skin, and I didn't care what others thought of me. I was taught never to let anyone tell me what I should or shouldn't wear, and while their taunts hurt, I refused to conform. I suppose now, I've sacrificed some of that comfort so I can keep jobs.*

One of Neve's main struggles in his life centers on his difficulties surrounding finding and/or maintaining a job. He states that, while he had worked "numerous jobs in sound engineering," his employment was "a constant roller coaster" due to his autism. At first, he did not want to apply for social security benefits, as he did not want to be considered a "disabled guy," but a conversation with his maternal grandfather reassured him that asking for help was not a shameful thing. In adulthood, Neve shares that he has now

*found other, more subtle ways of connecting with my inner self. I work with mixing consoles every day. I listen to music that speaks to my inner self. [...] I've found my niche, and my skill level at what I do far outweighs my social awkwardness enough that I can keep steady work.*

### 4.7. Endings: The Generativity Script

McAdams (1997, pp. 70, 71) states that the ending of a story should ideally bring about "new beginnings." This can be seen in the main threads of Neve's narrative. With regard to his interaction with the Otherkin community, Neve shares that he had made an attempt to connect with others again after being shunned in his teenage years:

*I stepped out of the community entirely by my senior year of high school. [...] I found a new forum where a lot of my old friends from back in the early days of my awakening had congregated. We were able to reconnect and I was welcomed there.*

However, due to full-time work and study at seminary, Neve "eventually became inactive." He left the Otherkin community, and started his page on DeviantArt, stating that he was still open for communication with any Machinekin who "need[ed] a understanding friend." As stated above, in 2016, after communicating with this author, Neve decided to make another attempt to connect with various (micro)communities, this time with success, most likely due to the fact that he had matured, and could clearly explain his own experiences for others to be able to understand. He also created his own website and Discord server, and has been able to build a community of his own. This online interaction, although uncertain and somewhat combative in the beginning, eventually transformed into a fruitful, productive means of expression for Neve.

With regard to his calling to the priesthood, his other jobs, and living situation, Neve also speaks of change. He tells of his realization that life as a parish priest was not for him, and that he decided he could "focus on the works of faith at home." In this same time period, he lost another job as a sound technician, and was "uprooted" due to loss of income; his mother fell ill with cancer and his great grandmother died; and he experienced his own health issues due to a loss of insurance and consequent lack of necessary medications, including hormone therapy. He "felt like crap every day" in this period; fortunately, circumstances began to improve for his mother and for himself as well. He was able to reconnect with his mother's family, and was able to obtain a home for himself. He feels he has been able to find many of the answers he has been seeking, but states,

*I'm nowhere near done learning. I've settled into being comfortable with myself and my living situation, but life has so much knowledge to offer [...] I've got a lot of years ahead of me to experience and grow in life.*

When asked how his faith has played a role in his life, and if religion has helped him find meaning for himself, Neve remarks that God purposefully created him, and strongly believes that his autism, his transgender and his Machinekin identities are all related to each other, as if by design: "They're all parts of me. I am one whole. Every aspect of myself intertwines to make up all that is me." Regarding his difficult life, he "absolutely" believes there is meaning to be found there as well: "I don't believe in coincidences. I've had every experience I've had because, like every soul, I've had things I needed to learn." For example, Neve feels the tragic death of his childhood friend spurred him to learn more about religion ("I knew, somewhere in my heart, there had to be something more"), the nature of God, the inherently good nature of human beings, and that he prefers to serve God in a monastic, prayerful fashion. He states, "That's my ministry. I put good energy out."

## 5. Discussion

The above analysis attempted to show how McAdams's life-story framework can be used to illustrate how narratives play an important role in the formation of a person's identity. In addition, examining narratives can demonstrate how telling a story about one's life events can also help one find meaning. In this case, religion is an important factor with regards to meaning-making in Neve's life. His belief that his human life is a miraculous gift that was purposefully designed by God solidifies for Neve his sense of who he is. With regard to this religious statement, we can refer to Zock (2013, p. 27), who discusses how a personalized religious tone can assist a person to "keep together a meaningful identity" and that this identity can be utilized "as a means of coping in major transitions in the life history." As Popp-Baier (2013, pp. 150–55) points out, narratives are a mixture of major and minor events that form coherence in a person's life, and as religion has become increasingly individualized, we can speak of a type of "individual religiousness" that calls for a narrative "as data," allowing scholars to illustrate "this personal religion with regard to biographical constellations." Some of the

themes that were gleaned from Neve's narrative are: *miraculous life, hardships, knowing self, purpose of life, serving God, need for connection, rejection, life changes, taking initiative, creativity,* and *kindness*; these can apply to Neve's everyday lived experiences and have contributed to his feeling of wholeness as a person.

We can also see how the use of qualitative research methods, including the use of narratives, can help scholars broaden the study of Otherkin, and prevent the study from being limited only to the categories of "religion" and "spirituality." For example, data taken from an independent, internal Otherkin survey (that includes both qualitative and quantitative data) from 2018 (see Appendix A of 196 participants demonstrate that the majority of respondents claim that: (1) the other-than-human identity is not a personal choice, although Davidsen (2014, pp. 258, 267–72, 274) and Kirby (2013, p. 56) suggest the identity is the result of a type of religious conversion; (2) the other-than-human identity is not dependent on a socially constructed community or worldview, as is claimed by Laycock (2012, p. 66); and (3) the majority of respondents feel a "religious" label is inaccurate, harmful, or misrepresentative of the larger community (Shea 2019, pp. 37–41, 52–54). These data suggest that a lacuna exists in the current research from Religious Studies scholars. By shifting focus to include the individual narratives of Otherkin, researchers can be better informed and can better understand the complex and diverse nature of individual experiences of Otherkin, that *can be* religious in nature for some individuals.

Additionally, in this case, we can see that the religious aspects of Neve's narrative fail to fully answer the question of *why* Neve identifies as other-than-human, i.e., the origins of this conviction. Other points surrounding the topic of identity come into question, such as: whether Neve's identity differs from other Machinekin; whether his narrative is unique as compared to other Machinekin narratives; whether a distinction exists between Neve's sense of his other-than-human self and his "humanness;" what the appeal is (if any exists) of this particular machine to Neve; whether Neve's autism influences his Machinekin identity; and whether or not gender is an important issue for Otherkin in general. This section will attempt to address these points in greater detail by drawing upon Neve's narratives, supplemented with this author's research data (including field notes) regarding how Otherkin think about themselves and their community.

This author's (unpublished) research from 2018 into aspects of inclusion and exclusion within the greater Otherkin online community demonstrates that two issues were at play with regard to exclusion: prejudice or discrimination against particular kintypes that were not seen as legitimate, and the need for strict boundaries to protect against what is known as "trolling" (i.e., causing harassment or discord) in online communities. Machinekin as a kintype was not readily accepted in the diverse micro-communities, although perceptions are currently changing. In addition to this, those people who purported to identify as objects (known by the pejorative term "objectkin") were also largely rejected, as objects were not considered to be capable of sentience. Neve's story is different from other Machinekin narratives in that Neve identifies as a music mixing console, as opposed to most other Machinekin who identify as robots/androids, or other forms of artificial intelligence. As Neve describes in his story (Section 4.3), objects such as mixing consoles were not seen as valid examples of possible Otherkin identities, and as he was young and not fully capable of convincingly articulating his thoughts about his identity, he was quickly seen as being a disruptive "troll" and was repeatedly banned. This author did not have interaction with any others within the community who identify as objects, so the opportunity to establish long-term communication with Neve has been notable.

As can be seen above, Neve believes in the concept of animism. In his narrative, we see that he feels he was born, or reincarnated, as the essence, or "soul" of a machine in a human body, and that this was decided by God to happen. He recognized his true identity through a powerful dream that was affirmed when he first found an image of the Neve mixing console online. The concept of animism within the Otherkin community is not unique, as there are other Machinekin (androids, for example) who also speak of having souls. What distinguishes Neve's narrative from others is the intervention of a divine power that determined the course of events. This particular point about the deliberate transmigration of a soul by a divine force has not been mentioned before to this author by

other Machinekin (but this author does not rule out the possibility that such narratives might exist). Most other narratives, this author was told, reflect a certain amount of "randomness" when referring to the process of the soul reincarnating into a human body.

Neve's conviction of his machine identity was seen (by him) not to be the result of obsessive behavior of an autistic person; on the contrary, the neurological differences of his autistic brain were *also* by design. God acted as the agent in his creation, and autism is thus seen as a necessary condition, and not a mistake. Neve speaks of his memories from his past life as the mixing console, suggesting that his machine "soul" was existing (by thinking, feeling, etc.) in another form, and another lifetime. In his case, God *planned* for Neve's soul to have a *human* experience, one that includes freedom of movement and other human sensory experiences (this is something that Neve had longed for in his previous life). Neve's identity is therefore not seen as a result of his extreme interest in music and music mixing consoles, something that might be assumed by his social environment (and something of which Neve is himself aware). To be clear, not all Otherkin are autistic, although this author interacted with several autistic Otherkin of varying types. However, the presumption that a link might exist between autism and other-than-human identities, and especially Machinekin identities, should be avoided at this time, as more research about autism and its relation to identity is needed in the greater Otherkin community regarding this sensitive issue. At present, this author can only state that autism plays an important role in Neve's narrative, and that it also is linked to the religious aspect of Neve's understanding of his identity. Furthermore, in learning more about how Neve thinks about the things that happen to him specifically, it could perhaps be fruitful to use the comparative phenomenology method, as is discussed by Luhrmann (2020) in her article, "Mind and Spirit: a comparative theory about representation of mind and the experience of spirit," in future research projects, as this might offer up new insights as to the various understandings of the mind and how these understandings might influence experiences that could be "deemed" spiritual or religious, as well as how the Otherkin subculture thinks about identity.

Neve's narrative presents a viewpoint of identity that is not static, but more fluid in nature, as is discussed by McAdams (1997) and Turkle (1997, 1999). His "real" self coexists with a human persona that the outside world witnesses; Neve had to find a way to reconcile these aspects, along with his transgender identity, for his own well-being and growth. Neve is aware that he is fully human in a biological sense, but explains that an "incongruence" exists between his mind/soul and body and that this has led to a desire to gain extra weight and, as already stated, to burden his physical body with large, heavy objects in order to add weight, girth, and width; therefore making his physical form appear angular and massive. He states that he is happy that his family are genetically predisposed to be overweight, and takes any comments about his large physical size to be a compliment and as something that is comforting to him (interview data from 2016).

What is particularly interesting in Neve's case is the following: (1) Neve's innate identity is other-than-human, but is located in a human body; he therefore must reconcile the feelings he has about his physical form that do not coincide with his mental/emotional representation of who he is; (2) the gender at birth (in this case, biological female) of the physical body that Neve "inhabits" does not correspond to his mental/emotional representation of his gender identity. As far as the data demonstrate, Neve speaks of knowing his Machinekin identity before any consideration of gender occurred, but this is not yet confirmed by this researcher, as the interviews primarily focused on the other-than-human identity. While Neve does state that he feels his Machinekin and transgender identities are connected, further research is necessary to determine to what degree, if any, a correlation exists between gender and other-than-human identities, not only in Neve's case, but also for the larger Otherkin community. This researcher communicated with others who also express themselves as being Otherkin and transgender, but this was not the case for all of the participants. Therefore, interdisciplinary research involving gender studies would be of significance in order to explore this topic in greater detail.

With regard to the question of whether or not the machine mixing board held a certain appeal to Neve, and the possibility of this appeal being the reason Neve identifies as a mixing board, this author's research demonstrates that the notions of appeal or affinity do not play a role for Otherkin. Many Otherkin speak of not liking their kintype at all, or not knowing anything about their kintype prior to realizing their other-than-human identity. As Otherkin explain it, it is something at is as innate as sexual orientation—you are simply born as you are, and there is really no answer that can be given as to *why* it is (Shea 2019, pp. 53, 54, 57, 58). Neve states above that he does not "know why God made [him] as He did," and this perhaps may be the only question that cannot be fully answered here. This, of course, does not mean that researchers cannot continue to explore this point of discussion. Again, insights from other areas of discourse, such as gender studies, cultural theory, and post-humanist studies, might be helpful when forming an interdisciplinary approach to this, and other questions. Otherkin think about their identities as more fluid in nature, and explore their identity (often through online interactions with like-minded people) in different ways. Otherkin are challenging boundaries of what being a person entails; we could perhaps speak of a new kind of person, as is discussed by Shane (2014). It would appear that differing ideas surrounding identity are becoming more prevalent in our society; the Internet, and (more broadly) media are areas where this is being explored. As scholars, it is important to be aware of, and recognize these shifts in thinking, taking these ideas into account when approaching further research in this new area, as well as in exploring the notion of experience. As researchers, we should perhaps not fall into a positivist/idealist trap when examining and interpreting the Otherkin experience, but remain methodologically agnostic, taking the position that other-than-human identity is something worthy of study. The importance of narrative should be apparent in this regard; people need to share the stories of their experiences in order for others to understand what is happening. This article shows an example of an alternate approach, and attempts to highlight the need for a change in research focus that concentrates on the subjective experiences of Otherkin identities. This can be realized through the use of qualitative psychological research methods, particularly the life-story interview, which demonstrates, in this case, how religion plays a role in the understanding of Neve's identity, and contributes to a sense of meaning for an individual.

**Funding:** This research received no external funding.

**Conflicts of Interest:** The author declares no conflict of interest.

**Ethical Statement:** All subjects gave their informed consent for inclusion before they participated in this research. The research for this article was performed in the context of research for this author's Research Master thesis; the approval of this author's research proposal included the ethical approval of this research.

## Appendix A. Survey of Otherkin/Therians' Feelings towards Representation of Themselves (Sample)

Note: "During the course of my research, I have met two Otherkin who act as helpful 'educators' of the community, and who have been conducting their own *emic* research for a book project. In May 2018, they conducted a six-month survey of Otherkin attitudes regarding a number of different quantitative and qualitative questions and responses. These include the academic interest in their community, and the material that has been written about them by scholars. These two Otherkin have been extremely supportive of my own research and have generously provided me the results of their recent survey of 196 participants. This survey was shared with 21 different Otherkin and Therian websites, groups, forums, discord servers, chats, social media platforms, etc." (Shea 2019, pp. 46, 47–51).

*May—November 2018 Authors: Nøkken and Hound*

| Do you consider yourself to be... | Percentage—196 responses |
|---|---|
| Otherkin | 17.3% |
| Therian (animal-only types) | 45.4% |
| Both Otherkin and Therian | 20.4% |
| Vampire | 8.2% |
| Did you consider yourself to be nonhuman prior to discovering the Otherkin/Therian community? | Percentage—194 responses |
| Yes | 73.7% |
| No | 7.2% |
| Unsure | 18% |
| Physically human, spiritually non-human | 0.5% |
| Physically human, spiritually/emotionally/mentally I've never felt human | 0.5% |

| What is your current age range? | Percentage—193 responses |
|---|---|
| 10–19 (years old) | 22.3% |
| 20–29 | 46.6% |
| 30–39 | 21.2% |
| 40–49 | 7.3% |
| 50–59 | 1% |
| 60+ | 1.6% |

| Do you believe that nonhuman identity was a personal choice? | Percentage—194 responses |
|---|---|
| No | 81.4% |
| Partially a choice, partially inherent part of me | 13.4% |
| Unsure | 3.6% |
| It was a choice to accept it | 0.5% |
| I made a choice to accept it when I learned more about it; also due to interaction in communities | 0.5% |
| It was a soul choice | 0.5% |

| Do you feel that your nonhuman identity is caused by atypical psychological experience and/or feelings? | Percentage—195 responses |
|---|---|
| Yes | 37.9% |
| No | 31.3% |
| Unsure | 27.2% |

| *Do you feel that your nonhuman identity is caused by atypical spiritual experiences?* | *Percentage—194 responses* |
|---|---|
| Yes | 50.5% |
| No | 25.3% |
| Unsure | 21.1% |
| *Do you consider Religious Studies scholars work about Otherkin/Therianthropy as being a religion or religious to be accurate?* | *Percentage—195 responses* |
| No | 64.4% |
| No, but it has religious elements | 26.7% |
| Unsure | 4.1% |
| Maybe, has yet to be determined | 3.1% |
| Yes, it is a new religious movement | 1.5% |

| *Do you feel that claims that Otherkin/Therianthropy is a religion or religious are harmful to the community?* | | | | | *1 = strongly disagree <-> 10 = strongly agree (5–6 neutral) 194 responses* | | | | |
|---|---|---|---|---|---|---|---|---|---|
| *1* | *2* | *3* | *4* | *5* | *6* | *7* | *8* | *9* | *10* |
| 4.1% | 1.5% | 6.7% | 3.1% | 9.8% | 9.8% | 12.9% | 11.9% | 8.2% | 32% |

| *Do you feel that academic claims that Otherkin/Therianthropy is a religion or religious are misrepresentations of the community?* | | | | | *1 = strongly disagree <-> 10 = strongly agree (5–6 neutral) 193 responses* | | | | |
|---|---|---|---|---|---|---|---|---|---|
| *1* | *2* | *3* | *4* | *5* | *6* | *7* | *8* | *9* | *10* |
| 1.6% | 0 | 0.5% | 0.5% | 6.2% | 6.2% | 7.8% | 14.5% | 13.5% | 49.2% |

| *Do you feel that Religious Studies scholars need more ethical oversight from the community itself when researching Otherkin/Therianthropy?* | | | | | *1 = strongly disagree <-> 10 = strongly agree (5–6 neutral) 194 responses* | | | | |
|---|---|---|---|---|---|---|---|---|---|
| *1* | *2* | *3* | *4* | *5* | *6* | *7* | *8* | *9* | *10* |
| 0 | 0.5% | 1% | 0.5% | 6.2% | 4.6% | 12.4% | 18% | 12.9% | 43.8% |

| *Do you feel that Otherkin/Therians have a right to determine whether they are considered a religion by others, including academics?* | | | | | *1 = strongly disagree <-> 10 = strongly agree (5–6 neutral) 192 responses* | | | | |
|---|---|---|---|---|---|---|---|---|---|
| 1 | 2 | 3 | 4 | 5 | 6 | 7 | 8 | 9 | 10 |
| 2.6% | 1% | 1.6% | 3.1% | 5.7% | 2.1% | 8.3% | 12% | 14.1% | 49.5% |

Note: The survey also provided space for qualitative answers regarding the questions about Religious Studies. Below is a selection (from the total of 57 responses) of qualitative answers regarding questions about Religious Studies:

- *A belief based on psychology, spirituality, and philosophy cannot be made a religion. As we have no set dogmatic scripture, practice, or anything else related to what defines religion in our community. We are a multi-cultural, multi-spiritual community made of way too many people to be so arrogant as to try and define it as something it is not.*
- *Therianthropy—and overall Otherkinity—is an identity. Period. Religion can have a part to play depending on the individual, but the identity itself is not a religion. It's like saying "LGBT+" is a religion. It's not. It's a part of someone's identity.*

- *The feeling of having a nonhuman identity is at the core of being otherkin. Whether the person involved attributes this identification to spiritual or psychological factors is a personal choice. We don't actually know what causes it. It is up to the individual to form their own beliefs and make their own conclusions. Belief in nonhuman past lives are common, but not universal. Such beliefs are not required to be otherkin.*
- *A religion requires a coherent set of teachings. While otherkinity has spiritual aspects for many of us, it isn't a religion because it's not organized in the same way a religion is.*
- *I have no idea how the idea of otherkin/therianthropy as a religion came about. It is quite clear within the community that none of us follow a specific religion or have a specific set of beliefs/rules/etc. Religion is utterly irrelevant to the otherkin/therian community; even the spiritual experiences some claim to have are almost always disconnected from religion/religious beliefs.*
- *Personally, I don't care what people believe or do. If they see their otherkinity/therianthropy as a religion, good for them. However, the identity itself is held by people of many religious backgrounds, and exists independently of it. There are also those who feel they are nonhuman because of psychology, which does not have any supernatural elements to it (which is a trademark of religion). There is really no central ideology to the community- the only thing that all of us share is that feeling of innermost nonhumanity.*
- *I'm a religious studies scholar, too, and so I have to tell that otherkin/therianism is much more like transsexualism. Both groups claim that they were born in the "wrong body"—but in therian/otherkin case the body belongs to a different species not a different sex/gender.*
- *I'm a completely atheist kin myself and I know there are others who are the same, so even ignoring how an identity cannot be classified as a religion, there are many non-religious kin in the first place. there are so many things wrong with trying to call it a religion and it's insulting* (Shea 2019, pp. 50, 51).

**Appendix B. Sample Transcript with Coding Analysis**

**Manual line-by-line coding [first table] and** *one degree axial coding [second table]* **PLUS theme(s) [in red] and "Life-story" feature [in blue]**
**Sample One:**

"I first discovered the Otherkin community in the early 2000s. One night, I decided to search "I do not feel human". This led me to a forum for Otherkin. The very first community I joined welcomed me. I made a lot of good friends over that forum and its corresponding IRC chat. At the time, I think I was the only machinekin known to the community. Some were skeptical and asked some prying questions, but overall, my experience with that particular group was positive. Thus, I sought out other communities. I found a very large forum for Otherkin, and my experiences there were just not the same. Bear in mind, I was a very young teenager, just a kid really—a lonely kid who couldn't put his thoughts together right. I was eventually banned from that forum under the belief I was nothing more than a troll. As time went on, it seemed the Otherkin community in general was becoming more closed off and defensive."

| |
|---|
| Discovered community in early 2000s; searched "I do not feel human", led to forum |
| First community welcomed me; made many friends. At that time likely only machinekin there. |
| Some skeptics, but overall positive experience. Sought out other communities. |
| Found large Otherkin forum, but different experience. Was a lonely, inarticulate kid then. |
| Was eventually banned from that forum; thought of as troll. Otherkin community seemed to become more closed-off and defensive. |

| |
|---|
| *Discovered community early 2000s with "I do not feel human" google search* |
| *Welcomed by first community; was likely only machinekin.* |
| *Some skeptics, but overall positive. Searched other communities.* |
| *Found another forum; different experience. Was lonely, inarticulate kid; was banned- seen as a troll.* |
| *Otherkin community became increasingly closed-off/defensive due to increasing trolling* |

Theme(s): other-than-human; searching for others; skepticism about machinekin; immature/inarticulate; discrimination; bullying; closed-off community

Life-Story feature: Theme

**Sample Two:**

"The first time I had any thoughts of spirituality, I was about ten. I had a best friend in the neighborhood who was disabled a lot like I was, and we got along. One day, a Friday afternoon, I was down at the corner park playing with my friend, and she invited me to come sleep over at her house. I went into the library there at the park, and used the payphone to call my mom. When I asked if I could go, my mom hesitated. She said, "You know, I don't think so. No. Come home." There was no particular reason for her to say no. It was a Friday night, I had no school the next day, and we had no plans as a family for Saturday morning. My friend only lived three blocks away. Begrudgingly, I walked home that night. The next morning, I woke up early and found my mom watching the news. That night, my friend's house blew up, and the entire family was killed. I could have been there, I could have died then, but miraculously lived."

| |
|---|
| Initial thoughts about spirituality around age ten; neighborhood best friend |
| Also disabled; playing at corner park with friend on a Friday |
| Invited to sleep over at friend's house; went to library to use |
| Payphone to call mom to ask for permission; mom hesitated, said no |
| Said to come home; no reason for refusal, no school the next day |
| No family plans; friend lived close by; reluctantly walked home |
| Next morning, mom watching news. Friend's house blew up, everyone died. |
| Could have been there, and could have died, but miraculously lived. |

| |
|---|
| *Initial thoughts about spirituality age ten* |
| *Played with neighborhood best friend, also disabled, at corner park on a Friday* |
| *Invited to sleep over by friend, went to library to call mother to ask permission; mom refused* |
| *Told Neve to come home, but no school or family plans the next day; reluctantly returned* |
| *Following morning on tv, news report of home explosion and deaths of friend and her family* |
| *Could have died but miraculously lived* |

Theme(s): spirituality, perplexed, loss of best friend, traumatic event, realization of miracle

"Life-Story" feature: Nuclear Episode (also, Theme—"miraculously lived")

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
