# Peer review of "Exploring Other-Than-Human Identity: Religious Experiences in the Life-Story of a Machinekin"

_religions, doi:10.3390/rel11070354_

Round 1
Reviewer 1 Report
Overall this is a well-written and fascinating examination of narrative identity formation based on thorough virtual ethnographic research buttressed with impressive theoretical scaffolding. I have three suggestions that I think would make the article stronger.
1) There needs to be a stronger overall guiding thesis to the article, an explanation not just of what will be done in the article but what the argument actually is. To my reading, the author wants to make an intervention in the way religion has been thought about with respect to Otherkin, moving away from the common analysis of Otherkin as religion to understanding how self-identified religious experience informs, or contributes to, identity making and narrativizing for Otherkin. The importance of this shift is that it aligns better with how most Otherkin talk about themselves and their experiences (see the very well written lines 78-84).
2) To help accomplish my first suggestion, I would move section 2, “Study Aims,” earlier in the article, before sections 1.1 and 1.2. Sections 1.1 and 1.2 are reading a bit too lit-review-ish. As a reader, I want to know earlier in the article what the author is going to actually be looking at, what the real meat of the article is. As written now, the author presents the intervention that he or she will be making before discussing the actual stuff of the intervention. It is more effective to say, “My research is on X, with Y conclusions that contribute in A, B, and C ways to ongoing discussions of Otherkin among Religious Studies scholars.” Section 1.2 could actually be better integrated, I think, into section 3, “Methods and Interview Information.”
3) I am unsure what section 5 is meant to do in this article. The author writes (page 10, lines 417-421):
Neve’s story is, however, an example of an unusual type of identity experience that can be discussed from different perspectives other than qualitative psychology and religion. For this article, four discussion points will be presented in the next two sections that relate to information taken from Neve’s narrative, found in diverse areas of study such as…
There is no doubt that Neve’s story of identity formation can be discussed from many different theoretical viewpoints and with relatedly different questions, but those seem like different articles. As written, section 5 reads like a sharp turn away from the careful McAdams-driven analysis of the previous section to just start something completely different. Section 5 also, to my reading, diverts from the author being able to provide a good conclusion for the analysis that he or she has actually done. Again, it is not that the discussion in section 5 is not important or interesting, I am just unsure how it works in this specific article for the journal Religions.
All of my suggestions can be summed up, I think, with: this is really interesting and worthwhile work, but the focus needs to be tighter on the single contribution of the article.
Reviewer 2 Report
This is a very interesting contribution aimed at describing certain conceptual aspects (esp. identity, personhood in connection with religiosity, spirituality) of a novel and understudied WWW phenomenon (Otherkin/Machinekin). The author's argument is compelling, the research is competent and extensive, and the research methods and results are explained clearly.
I have three concerns: (1) Does the Machinekin who identifies himself as Neve know that this research focused on his life story and personal identity is being published? That's my only ethical concern, i.e. that Neve, who is the subject of this research, may not appreciate being discussed in a public academic forum.
(2) I wonder if the author can explain the different forms that questions about identity can take. For instance, one question is what makes this particular Machinekin unique. In other words, how is he different from other Machinekin individuals, and has he given any thought to this unique aspect of his identity? Another question is what makes Neve a person as distinct from what makes him human? In general, when we talk about identity in Philosophy, we aim to differentiate the question about personhood from that about identity
(3) It is not clear from the interviews with Neve why he identifies with a sound mixing console in particular. Why that type of machine specifically? And why machine in the first place? I mean, what is it about a machine that appeals so much to Neve? Is it automaticity? Programability? Predictability? The artificial intelligence aspect of machines? There may be an implicit psychological connection here between aspects of autism and the features of artificial intelligence. Finally, the author says nothing about autism and identity, and there have been several studies devoted to the sense of self autistic individuals develop.
